# The Role of Radiomics in Fibrosis Crohn’s Disease: A Review

**DOI:** 10.3390/diagnostics13091623

**Published:** 2023-05-04

**Authors:** Ali S. Alyami

**Affiliations:** Department of Diagnostic Radiography Technology, College of Applied Medical Sciences, Jazan University, Jazan 45142, Saudi Arabia; aalmansour@jazanu.edu.sa; Tel.: +966-551234818

**Keywords:** IBD, Crohn’s disease, radiomics, fibrosis, imaging

## Abstract

Inflammatory bowel disease (IBD) is a global health concern that has been on the rise in recent years. In addition, imaging is the established method of care for detecting, diagnosing, planning treatment, and monitoring the progression of IBD. While conventional imaging techniques are limited in their ability to provide comprehensive information, cross-sectional imaging plays a crucial role in the clinical management of IBD. However, accurately characterizing, detecting, and monitoring fibrosis in Crohn’s disease remains a challenging task for clinicians. Recent advances in artificial intelligence technology, machine learning, computational power, and radiomic emergence have enabled the automated evaluation of medical images to generate prognostic biomarkers and quantitative diagnostics. Radiomics analysis can be achieved via deep learning algorithms or by extracting handcrafted radiomics features. As radiomic features capture pathophysiological and biological data, these quantitative radiomic features have been shown to offer accurate and rapid non-invasive tools for IBD diagnostics, treatment response monitoring, and prognosis. For these reasons, the present review aims to provide a comprehensive review of the emerging radiomics methods in intestinal fibrosis research that are highlighted and discussed in terms of challenges and advantages.

## 1. Introduction

In medicine, medical imaging is a basic technology, and in clinical practice, it is used to support decision-making for screening, diagnostic, therapeutic, and follow-up purposes [1,2]. Conventional imaging modality evaluation relies heavily on subjective interpretation by radiologists, increasing the possibility of interpretive variability. These modalities, such as ultrasound (US), magnetic resonance imaging (MRI), computed tomography (CT), etc., recognize images of internal organ structures and aid in the evaluation of physiological function and pathology change in human systems.

Artificial intelligence, deep learning, computer vision, and radiomics have the potential to revolutionize almost every field that relies on cross-sectional medical imaging for diagnosis and follow-up, including IBD. Radiomics is the process of converting digital medical images into mineable, high-dimensional data. It includes the computational extraction of shape, intensity, and appearance descriptors from medical imaging, and when integrated into ML algorithms, it can predict outcomes of interest. Radiomics is motivated by the idea that biomedical images contain information that reflects underlying pathophysiology and that these relationships can be revealed through quantitative image analyses. It is intended to develop decision support tools; therefore, it entails combining radiomic data with other patient characteristics, when available, to enhance the predictive power of the decision support models. It is non-invasive and reproducible, offers diagnosis and stages of cancer, helps in therapy planning, and predicts therapeutic outcomes [3].

In the field of oncology, radiomics has gained notable traction. For instance, authors trained and validated a radiomics-driven machine learning (ML) model to predict the histopathologic response to neoadjuvant chemoradiation using baseline T2-MRI images of primary rectal cancers [4]. Recently, clinical uses of radiomics in Crohn’s disease (CD) have started to be investigated. Using this advanced technique to improve intestine segmentation techniques for cross-sectional imaging has the potential to reduce variability and improve disease activity measures. For example, radiologists could benefit from automation to score staging more objectively, quantify bowel features, and reduce interobserver variability. In IBD, radiomics represent a promising tool that could assist radiologists in better assessing disease severity, clinical decision-making, and patient phenotyping. Indeed, through texture analysis, numerous quantitative pieces of information may be retrieved from radiological scans and placed into machine learning models to predict desired outcomes [5]. Radiomics has proven to be a useful technique in the gastrointestinal field, allowing radiologists to be more accurate in predicting gene signatures [6], diagnosing rectal cancer, colon, small bowel, and other lesions [7,8,9], detecting fibrosis in paediatric CD patients [10], and evaluating treatment response [7,11]. However, only a few studies have focused on CD diagnosis [12]. In this context, the researchers focused primarily on how radiomics could aid in CD diagnosis [13], evaluation of therapeutic response [14,15], and severity of disease and its complications [16,17,18].

Radiomics has been applied to different intestinal diseases, not only CD, such as colorectal cancer, and intestinal lung disease, such as pulmonary fibrosis. Recent research has shown promising results in the use of radiomics at various stages of colorectal cancer diagnosis and treatment, including preoperative, intraoperative, and postoperative phases [19]. In the field of pulmonary fibrosis, radiomics have been used with many benefits, such as differentiating between different types of pulmonary fibrosis [20], predicting the functional decline and long-term survival of pulmonary fibrosis patients [21], and enhancing the diagnosis and severity staging of idiopathic pulmonary fibrosis [22]. The added utility of radiomic characteristics in the differential diagnosis of ulcerative colitis and CD has been assessed. Li et al., for this purpose, constructed a nomogram based on CT radiomics features mixed with clinical factors, with an area under the curve (AUC) on the test set of 0.8846 [13]. Radiomics has recently proven to be a valuable tool not only for supporting radiologists in the staging and diagnosis of CD but also in the evaluation of the response to therapy. In terms of MRE studies, some authors developed and validated an MRI-based radiomics nomogram for detecting secondary loss response to infliximab in a group of CD patients [14], while others built several MR enterography (MRE) radiomics-based machine learning models for predicting response to immunosuppressive treatment in patients with CD, reaching AUCs ranging from 0.71 to 0.99 [15]. On computed tomography enterography (CTE) examinations, other researchers developed a CTE-based radiomics nomogram to predict loss of response to infliximab in CD patients, which demonstrated good performance and a subsequent clinical benefit [14]. The assessment of disease severity at imaging is critical for patient management, but it is limited by inter-reader variability. On MRI examination, Ding et al. attempted a more reproducible and objective approach by stratifying CD severity in the terminal ileum via the extraction of radiomics, achieving comparable outcomes to MaRIA values scored by a senior radiologist in their most recent study [16]. Kurowski et al. observed that the heterogeneity of radiomic characteristics of visceral adipose tissue is greater in CD patients than in controls, which may be an indicator of disease severity in paediatric CD patients [18]. Moreover, radiomics features can then be used to develop predictive models for fibrosis in CD. Fibrosis in CD is a chronic inflammatory condition that can lead to intestinal strictures, obstruction, and a bowel perforation. The ability to accurately predict the development and progression of fibrosis in CD is critical for identifying patients who may benefit from early intervention. However, quantifying active inflammation versus fibrosis is difficult; in fact, no technique is accurate enough to assess the degree of fibrosis in stricture and guide clinical choices. Radiomics has the potential to provide valuable insights into the development and progression of fibrosis in CD. By using advanced imaging techniques such as MRI or CT scans, radiomic features can be extracted from the images and used to develop predictive models. This review will review the previous literature using the radiomics of CT and MRI modalities and discuss the radiomics of the advantages and disadvantages of fibrosis CD in general.

## 2. Radiomics Workflow

In brief, the radiomics procedure is divided into different stages of acquiring the image and segmenting the volume, feature extraction and storage, and signature development and validation on one or more datasets, as seen in Figure 1. Each of these three steps presents its own set of obstacles. The process is the same once a signature has been developed and applied to a specific patient, except that in stage three, the validated signature is used to determine the patient’s prognosis.

### 2.1. Image Acquisition

The first step in the radiomics workflow is acquiring the medical images, which can be CT or MRI in the case of fibrosis in CD. Modern versions of these scanners support a wide variety of acquisition and reconstruction settings. Although this facilitates the subjective demands of the human expert, when the images are supposed to be objectively defined by a machine, these differences may produce a bias that masks the true underlying biological characteristics. In the field of radiomics, this phenomenon is well known, and attempts are being made to reconstruct techniques and standardize acquisition.

### 2.2. Regions of Interest

On diagnostic imaging, identifying one or more volumes of interest is one of the central processes of radiomics. Predictive value, however, may be found in the detailed analysis of tumor sub-volumes.

### 2.3. Segmentation

The next step is to segment the images to isolate the region of interest (ROI), which is typically the area affected by fibrosis in CD. The segmentation of ROI could be divided into semi-automatic/automatic, and manual segments. Delineation, or segmentation, plays a critical role in radionics because the features that are generated depend on the segmented volumes. However, the borders of numerous tumors and sub-volumes are unclear. When these volumes are manually delineated, this can lead to substantial inter-reader bias and low reproducibility. Unfortunately, no universal automatic segmentation algorithm exists that can be applied to all medical modalities [23]. A consensus arises from this debate that optimal reproducible segmentation can be achieved through semi-automatic segmentation, which comprises automatic segmentation followed by manual curation if necessary [24]. Additionally, each imaging modality has its own unique segmentation technique. For example, in PET, the metabolic target volume (MTV) is segmented as an ROI, while in CT, the ROIs represent the gross tumor volume (GTV).

### 2.4. Feature Extraction

The high-throughput extraction of quantitative image characteristics that characterize the ROI is a critical component of radiomics. There is a risk of overfitting because of the large and intricate number of features that exceeds a thousand. To prevent this, the ratio of evaluated features to outcome occurrences must be as small as possible. A major technique for mitigating this risk is the reduction and ranking of features.

### 2.5. Model Development and Validation

The development of a model based on calculated radiomic features might be hypothesis-driven or data-driven. The hypothesis-driven method treats cluster features according to clinical and predefined information contents, while the data-driven method makes no assumptions about the significance of individual characteristics; hence, all characteristics are given equal weight during model development. The best models begin with a well-defined endpoint, such as overall survival, and integrate non-radiomics features whenever possible. Frequently, model performance is evaluated based on calibration and discrimination. Models that are accurate and correct differentiate between patients. For censored data, this can be measured using the AUC of the receiver operating characteristic (ROC) or the c-index [25]. Calibration, on the other hand, is the relationship between model prediction and observed outcomes.

## 3. Analysis of the Literature

A review of all literature on using radiomics in intestinal fibrosis in patients with CD demonstrates that radiomics are effective in terms of accurate disease detection and classification of intestinal fibrosis in CD using MRI and CT, as seen in Table 1. When it comes to radiomics analysis, both CT and MRI can provide valuable quantitative features for analysis. However, the specific features that are extracted may differ depending on the imaging modality and the specific radiomics approach used. Correlating models obtained from imaging features with histological findings is one area where radiomics research can expand.

### Application of Radiomic in CT in Intestinal Fibrosis

CT scans are often preferred for their speed and availability, and they are particularly useful for evaluating the extent and severity of fibrosis in the gastrointestinal tract. CTE is one of the most effective and common modalities for detecting and monitoring bowel disease in CD patients [27]. Previous studies have, however, reported that conventional CTE findings examined by radiologists did not correlate with intestinal fibrosis, suggesting that CTE is ineffective for assessing intestinal fibrosis [28]. The combination of CT scan radiomics and fibrosis assessment in CD is an exciting area of research that has the potential to improve the accuracy and precision of fibrosis diagnosis and monitoring. Radiomics features extracted from CT have been used to distinguish CD from intestinal tuberculosis [29], to distinguish CD from ulcerative colitis [13], to measure intestinal fibrosis in CD [17,26], and to predict the loss of the secondary response to infliximab in CD [14].

A few studies used radiomics in IBD in the recent retrospective study by Stidham et al. CTE developed a semi-automated quantitative measurement of bowel features such as lumen diameter, bowel wall thickness, and dilation. They found a very good maximum bowel wall thickness association between the mean measurement done by two radiologists and the semiautomated way (r = 0.702), which included 138 scans [12]. Another study in which the radiomics model was seen to outperform radiologists was a multicenter, retrospective CTE that included 167 CD patients with 212 bowel lesions. The machine learning-based radiomic model has high accuracy for predicting the presence of intestinal fibrosis, with AUROCs of 0.724–0.816 compared to radiologists’ AUROCs of 0.554–0.556 [17]. The study found that the radiomic model had excellent predictive accuracy for identifying the presence of fibrosis, with an area under the curve (AUC) of 0.92 in the validation cohort. The radiomic model was also able to identify specific radiomic features that were associated with fibrosis, including texture features related to heterogeneity and contrast. They concluded that the CTE-based radiomics model performed excellently in diagnosing intestinal fibrosis in patients with CD, but this radiomic model was too time-consuming to delineate the volumes of each damaged bowel segment on CTE. Furthermore, because radiomic features are handcrafted, the radiomic model is subjective. Moreover, further studies are needed to validate these findings and develop standardized radiomic protocols for clinical use.

Deep learning is an automated method that is less reliant on human involvement, which can significantly reduce the time required for identifying the volumes of interest (VOI) and extracting and selecting features. It can also reduce subjectivity by using a convolutional neural network for feature extraction, which is a data-driven approach [30]. A recent study by Meng et al. developed and validated a CTE-based deep learning model for characterizing bowel fibrosis in 235 patients with CD in a retrospective study. They compared deep learning and radiomics models and radiologists evaluations for intestinal fibrosis in CD-affected patients. They found this model can accurately distinguish moderate-to-severe from non-mild intestinal fibrosis in patients with CD. Furthermore, its performance was superior to that of the radiologists and was not inferior to that of the radiomic model, with a much shorter processing time, suggesting that this model may help radiologists grade bowel fibrosis more quickly and accurately [26].

#### Application of Radiomics in MRI

Although both conventional multi-phase contrast-enhanced MRI and CT can be utilized to get the distinct image features of CD, MRI offers several additional imaging sequences to help in the diagnosis of CD without causing radiation damage. Moreover, it has the ability to observe the bowel transmurally and from different perspectives. Furthermore, the combination of MRI parameters derived from different conventional MR sequences from magnetization transfer imaging (MT) (MT ratio), native T1 mapping, and diffusion weighted image DWI (apparent diffusion coefficient; ADC) provides a contribution to the detection of intestinal fibrosis. MT is a type of MR sequence that delivers a continuous measurement (MTR, 0–100%) generated by dipolar and generates contrast by interacting between protons in large stationary macromolecules and those in free water, such as collagen. The interaction between water and collagen in tissues results in the saturation of water magnetization and subsequent signal intensity loss due to magnetic and exchange couplings. As the proportion of collagen in tissue increases, there is a corresponding increase in signal loss and the MT effect. The MTR signal increases as the amount of macromolecules (including collagen) increases, and it should potentially be able to identify highly fibrotic tissues [31]. Initial studies indicate that MT-MRI can be utilized to identify bowel wall fibrosis and differentiate between inflamed non-fibrotic and inflamed fibrotic segments [32,33]. A study conducted by Fang et al. demonstrated that combining MT-MRI with conventional MRI can enhance the ability to differentiate between fibrotic and inflammatory components of small bowel strictures. The study found a significant correlation between histological fibrosis scores and MTR (*p* < 0.001), enabling the detection of mild versus moderate-to-severe fibrosis with a high sensitivity of 91% and specificity of 92% [32]. Native T1 mapping is a quantitative method for identifying fibrotic features in CD. Both native T1 mapping and MTI have been established as promising advances in the area of MRI in terms of detecting and distinguishing bowel fibrosis [34].

Additionally, ADC has been found to be significantly correlated with histopathologically derived fibrosis and inflammation scores, as well as percent gain. Further, based on an established cutoff value, the ADC was able to accurately distinguish fibrosis in CD with a sensitivity of 72% and a specificity of 94%. This suggests that ADC has the potential to be a useful non-invasive technology for identifying fibrosis in Crohn’s disease [35]. Another study found that there was a significant correlation between ADC and histologically derived inflammation grades in CD. Additionally, the study showed that both ADC and apparent diffusional kurtosis were significantly correlated with histologically derived fibrosis grades. The study also demonstrated that apparent diffusional kurtosis was able to accurately distinguish between the absence of fibrosis or mild fibrosis and moderate to severe fibrosis in Crohn’s disease with a high sensitivity of 95.9% and specificity of 78.1%. This suggests that apparent diffusional kurtosis has the potential to be a useful tool in assessing bowel fibrosis using MRI imaging [36]. Additionally, the aforementioned MSOT imaging technology has the potential for future intestinal fibrosis diagnostics due to its capability to detect collagen as a result of the exhibited optoacoustic signal [37].

Radiomic features have been shown to quantify subtle details on MRI examinations that may not be visible visually and that can be correlated with underlying pathophysiology [38] and treatment response [39,40].

In CD, the initial studies of radiomic features have shown that they may be able to characterize bowel fibrosis [17] and capture histological disease activity [41,42], suggesting their potential to prognosticate the need for surgery in CD. Previous MRI studies in CD have stated that diseased bowel segmentation with MRI using a convolutional neural network or active learning takes about 12 min [43]. One retrospective study conducted in a rat model showed that MRI-based deep learning allowed the evaluation of intestinal fibrosis in IBD [44]. This study included 45 rats’ models of inflammation (35 irradiated with visible lesions and 10 controls). It was found that this method provides practitioners with a useful tool for evaluating antifibrotic treatments in the development and extrapolating of such non-invasive MRI score models for patients with the goal of identifying early stages of fibrosis and improving disease management [44].

Another study focused on the use of texture analysis in intestinal fibrosis among 25 CD paediatric patients. The radiomic characteristics (texture analysis) of MRE contrast enhancement have been utilized to detect fibrosis in the bowel strictures of paediatric patients with CD [10]. This study showed that texture entropy can distinguish strictures with fibrosis (mixed or purely fibrotic) from inflammatory strictures. In addition, they found that a computer had superior diagnostic accuracy than a human radiologist [10]. The computer differentiated mild or no fibrosis from moderate to severe fibrosis with an AUC of 0.995 using histology as a gold standard. Lamash et al. conducted a retrospective cohort study among 23 active CD patients using semi-supervised and active learning models vs. convolutional neural networks. They found that convolutional neural network segmentation of the bowel’s background, wall, and lumen was compared with manual boundary delineation. CNN segmentation of dynamic contrast enhancement showed high agreement with manually segmented bowel images in 79%, 81%, and 75% of cases, respectively. The median value of relative contrast enhancement (*p* = 0.0033) and extracted markers of wall thickness at the location of min radius (*p* = 0.0013) could distinguish nonactive and active disease segments. Other retrieved markers could distinguish between segments with and without strictures [43].

There is potential to correlate models obtained from the imaging features extracted with histological findings in IBD, provided that segmentation methods of the bowel wall are robust to nonrigid motion, as typically seen in the bowel. One study aimed to investigate whether MRI texture analysis (MRTA) of T2-weighted images could provide information on histological and MRI disease activity in patients with Crohn’s disease undergoing ileal resection. Makanyanga et al. reported that MRI CD activity scores and histological measures of CD activity were associated with MRI texture features in a pilot study that included 16 CD patients who underwent MRI and ileal resection [41]. The study found that MRTA was able to accurately predict histological and MRI disease activity scores with high accuracy. The researchers also identified specific texture features that were associated with disease activity, including entropy, contrast, and homogeneity. In another retrospective, the multicenter study demonstrated that a radiomic model derived from MRE and CTE data detected moderate-to-severe fibrosis in the intestine with diagnostic accuracy comparable to radiologist assessments [45]. While several studies have shown the potential of MRI radiomics in assessing fibrosis in CD, further validation is needed to determine its clinical utility and ability to predict treatment response and outcomes.

## 4. Radiomics Limitations

As a new and developing medical tool, radiomics is bound to face difficulties, limiting its widespread use in clinical practice. These pitfalls can be found in the majority of the radiomic workflow steps. One limitation of radiomics is the transferability and reproducibility of radiomics features, as they are heavily dependent on the type of modalities, parameters, intensity normalization, segmentation, sequences, resolution, co-registration, acquisition protocols, quality, size, and motion artifacts of image transfer. For instance, there are differences in section thickness between the different modalities and imaging centers, dose administration, and reconstruction kernels.

The dependent correlation between the features and the input data, as the features are generated from that very database, is another limitation of radiomics. As a result, large datasets are required, in contrast to feature-based radiomics, to accurately identify relevant and robust feature subsets. Moreover, manual segmentation methods have inherent intra- and inter-observer variability, which could be reduced by using semi- or fully automatic techniques.

The radiomics have some technical challenges. CT and MRI imaging may require specialized training and expertise, as well as access to specialized equipment and software. This can limit its availability and feasibility in some settings. In addition, many studies using these radiomic modalities in CD have been limited by small sample sizes, making it difficult to draw firm conclusions and generalize the findings to a larger population.

## 5. Future Directions

In general, using radiomics in CD faces numerous challenges before it can be used in a daily clinical setting. These obstacles, including its technical complexity, standardization of data analysis, and acquisition protocols, are required to provide a robust framework. Another obstacle is that, due to the success of the procedure depending on the operator’s expertise, manual segmentation of the target ROI may result in higher interobserver variability and lower efficiency. In addition, there is a need for better accuracy in automatic segmentation. Furthermore, the enhancement of reproducibility is influenced by a number of factors, including image reconstruction, acquisition, and analysis.

A better understanding of the pathophysiological interaction between muscular hyperplasia, inflammation, fibrosis, and overall stricture narrowing is required. To reliably validate cross-sectional imaging techniques that guide CD-stricture medical and surgical care, validated histopathological index scores are essential. The heterogeneity of histomorphology within strictures, as well as the fact that most internal penetrating disease coexists with strictures [46], make cross-sectional imaging modalities difficult to detect these distinct elements. Addressing these issues would significantly contribute to reshaping the scope of CD care, particularly with anti-fibrotic therapies on the horizon.

Although cross-sectional imaging modalities such as MRE and CTE demonstrate excellent performance in detecting strictures, MRI-based modalities may offer promising new parameters for distinguishing inflammation from fibrostenosis. Upcoming modalities, such as the Type I Collagen-Targeted MRI Probe, may improve the ability to stage fibrosis in CD [47].

## 6. Conclusions

In CD patients, advanced imaging plays an essential role in the diagnosis, staging of lesions, characterization of disease consequences, and mentoring treatment response. In this context, current advances in radiomics can assist physicians in enhancing diagnostic accuracy and stratifying patients based on their prognosis in the direction of a customized medicine strategy. Fibrosis is a common pathophysiological result of chronic inflammation. Additionally, radiomics may have great potential to enhance the diagnosis and classification of fibrosis in patients with CD. Furthermore, to improve the feasibility of radiomics clinical applications, a more standardized methodology in the radiomics workflow is required, particularly in terms of study design and validation to differentiate between fibrosis and inflammation.

## Figures and Tables

**Figure 1 diagnostics-13-01623-f001:**
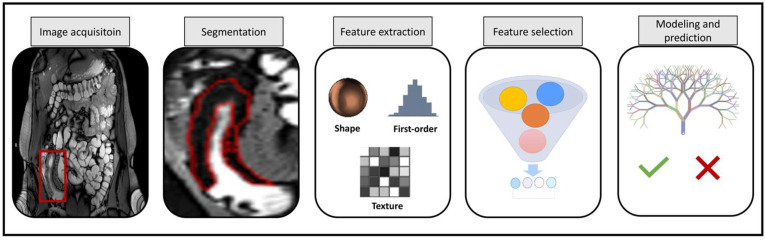
Shows general radiomics steps for CD classification from MRI images.

**Table 1 diagnostics-13-01623-t001:** A brief sample of literature studies.

Authors	Aim	Number of Patients and Type of Study	Gold Standard	Imaging	Endpoint
Meng et al. [26]	Comparison of deep learning and radiomics models, and radiologists’ evaluation for the of intestinal fibrosis in patients with CD	235 patients (312 bowel segments) Multicentric, retrospective	Histology	CTE	In diagnosing intestinal fibrosis in patients with CD using CTE, deep learning mode is better than radiologists and not inferior to the radiomics model.
Li et al. [17]	Developed a CTE-based radiomic model for characterizing intestinal fibrosis in patients with CD	167 CD patients 212 bowel lesions (test 114 lesions; training 98 lesions) Multicentric, retrospective	Histology	CTE	CTE-based RM allows for the accurate characterization of intestinal fibrosis in CD. The radiomic model achieved an AUC between 0.724 and 0.816, which significantly outperformed radiologist-interpreted imaging signs AUCs <0.600 in the evaluation of the severity of intestinal fibrosis in CD
Tabari et al. [10]	Evaluate if texture analysis of contrast enhanced MRE images can determine CD stricture histologic type.	25 CD pediatric patients Monocentric, retrospective	Histology (bowel resection)	MRE	The goodness-of-fit AUC for texture analysis of bowel wall signal intensities for detecting stricture fibrosis was 0.995.

## Data Availability

The dataset used during the current study is available from the corresponding author upon reasonable request.

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
