# Peer review of "The Role of Radiomics in Fibrosis Crohn’s Disease: A Review"

_diagnostics, 2023, doi:10.3390/diagnostics13091623_

Round 1
Reviewer 1 Report
In the present review article Alyami discussed about possible applications of radiomics in patients with Crohn’s disease (CD) to detect intestinal fibrosis. The article is simple, clear and easy to read. I have only one comment. I assume that the gold standard for fibrosis confirmation was histology (on surgical specimens). Authors should therefore specify this detail and add a column in Table 1 stating the gold standard for each study.
None
Author Response
I thank this reviewer for their comments, and I agree with the reviewer that the gold standard for fibrosis confirmation is histology (on surgical specimens). Thus, I have specified this detail and added a column in Table 1 stating the gold standard for each study, as requested. Please page 8 highlighted in red.

Reviewer 2 Report
The author reviews the emerging radiomics methods in intestinal fibrosis research, highlighting and discussing their challenges and advantages. It is quite an interesting article.
The only comment:
Figure 1: has the author a permission for using this image?
Author Response
I thank this reviewer for their comments. I have designed this image by myself. Yes, it may look like other ones in the literature, but this is my own one.

Reviewer 3 Report
The author reviews the role of Radiomics in fibrosing Crohn's disease (CD).
We are now in the dawn of an era in which AI and other technologies for image analysis do not need to be based on the reader's intuition or experience, but only on the AI's learning of results that surpass them.
There seems to be no problem with what is described, etc.
However, if I were to express a desire, I would like to know, for example, how much technological progress has been made in the evaluation of fibrosis using Radiomics, and how does it compare with CD in the field of pulmonary fibrosis, for example? It would be even better if there were a brief description of the function of radiomics compared to other intestinal diseases, such as colorectal cancer or pseudomembranous enteritis.
Author Response
I thank this reviewer for their comments, and I have addressed the comments by the reviewer in the text. I have added a bit more information about the role of radiomics in other diseases, as requested. Please see the introduction, page 2 highlighted in red.
